# Tumor Necrosis Factor Alpha: Implications of Anesthesia on Cancers

**DOI:** 10.3390/cancers15030739

**Published:** 2023-01-25

**Authors:** Wei-Cheng Tseng, Hou-Chuan Lai, Yi-Hsuan Huang, Shun-Ming Chan, Zhi-Fu Wu

**Affiliations:** 1Department of Anesthesiology, Tri-Service General Hospital, National Defense Medical Center, Taipei 114, Taiwan; 2Department of Anesthesiology, Kaohsiung Medical University Hospital, Kaohsiung Medical University, Kaohsiung 807, Taiwan; 3Department of Anesthesiology, Faculty of Medicine, College of Medicine, Kaohsiung Medical University, Kaohsiung 807, Taiwan; 4Center for Regional Anesthesia and Pain Medicine, Wan Fang Hospital, Taipei Medical University, Taipei 116, Taiwan

**Keywords:** anesthesia, cancer surgery, tumor microenvironment, tumor necrosis factor alpha

## Abstract

**Simple Summary:**

Cancer ranks as a leading cause of death worldwide, which is often the result of recurrence and metastasis. Surgical resection is the mainstay of cancer treatment for potentially removable solid tumors. However, surgery-induced stress responses may lead to immunosuppression and subsequent cancer regrowth and spread. Evidence shows that the tumor microenvironment plays an essential role in disease progression through mechanisms such as inflammation promotion. Tumor necrosis factor alpha (TNF-α) is one of the pro-inflammatory cytokines found in cancer patients and is reported to be involved in the immune system as well as in the surveillance of tumor growth. To date, several studies have demonstrated that various anesthetic agents or techniques and perioperative management have varying effects on innate and cellular immunity, the enhancement of adrenergic-inflammatory responses, and the activation of cancer-promoting cellular signaling pathways, which may be associated with long-term cancer outcomes. This review outlines the current knowledge of anesthesia implications on TNF-α release and provides potential anesthetic strategies for improving patient survival.

**Abstract:**

Cancer remains a major public health issue and a leading cause of death worldwide. Despite advancements in chemotherapy, radiation therapy, and immunotherapy, surgery is the mainstay of cancer treatment for solid tumors. However, tumor cells are known to disseminate into the vascular and lymphatic systems during surgical manipulation. Additionally, surgery-induced stress responses can produce an immunosuppressive environment that is favorable for cancer relapse. Up to 90% of cancer-related deaths are the result of metastatic disease after surgical resection. Emerging evidence shows that the interactions between tumor cells and the tumor microenvironment (TME) not only play decisive roles in tumor initiation, progression, and metastasis but also have profound effects on therapeutic efficacy. Tumor necrosis factor alpha (TNF-α), a pleiotropic cytokine contributing to both physiological and pathological processes, is one of the main mediators of inflammation-associated carcinogenesis in the TME. Because TNF-α signaling may modulate the course of cancer, it can be therapeutically targeted to ameliorate clinical outcomes. As the incidence of cancer continues to grow, approximately 80% of cancer patients require anesthesia during cancer care for diagnostic, therapeutic, or palliative procedures, and over 60% of cancer patients receive anesthesia for primary surgical resection. Numerous studies have demonstrated that perioperative management, including surgical manipulation, anesthetics/analgesics, and other supportive care, may alter the TME and cancer progression by affecting inflammatory or immune responses during cancer surgery, but the literature about the impact of anesthesia on the TNF-α production and cancer progression is limited. Therefore, this review summarizes the current knowledge of the implications of anesthesia on cancers from the insights of TNF-α release and provides future anesthetic strategies for improving oncological survival.

## 1. Introduction

Cancer is a major public health issue and a leading cause of death worldwide, accounting for 19.3 million new cases and almost 10.0 million deaths in 2020 [1]. Despite advancements in chemotherapy, radiation therapy, and immunotherapy, surgery is the mainstay of cancer treatment for solid tumors [2,3]. However, up to 90% of cancer-related deaths are the result of metastatic disease after surgical resection [4]. Tumor cells are known to disseminate into the vascular and lymphatic systems during surgical manipulation and subsequently migrate to distant organs and initiate tumor regrowth [5,6]. In addition, surgery-induced stress responses can lead to the immunosuppression and upregulation of adhesion molecules through mechanisms involving inflammation, ischemia–reperfusion injury, sympathetic nervous system activation, and increased cytokine release, significantly increasing the risk of metastatic recurrence [5]. Ultimately, the combination of potential tumor cell dissemination and an immunosuppressive state may result in the growth of residual cancer and metastatic seeding. Thus, modulating the cellular and molecular profiles that contribute to tumor recurrence and metastasis during surgery is critical to improve the outcomes for cancer patients (Figure 1).

The tumor microenvironment (TME) is the cellular environment in which the tumor exists and is composed of various cell types (fibroblasts, endothelial cells, immune cells, etc.) and extracellular components (cytokines, growth factors, extracellular matrix, etc.) [7]. The interactions between tumor cells and the TME not only play decisive roles in tumor initiation, progression, and metastasis but also have profound effects on therapeutic efficacy [8]. Evidence shows that the TME can harbor tumor cells to invade healthy tissues and spread to other body parts through the lymphatic or circulatory system; however, non-malignant cells in the TME perform a pro-tumorigenic function in all stages of carcinogenesis by stimulating and facilitating uncontrolled cell proliferation [9]. Notably, surgery itself and the resultant stress responses have direct impacts on tumor cells and can form a supportive TME through platelet activation, neutrophil extracellular traps, and immunosuppression, promoting occult tumor growth and the metastatic process [10,11].

Tumor necrosis factor alpha (TNF-α) is a pleiotropic cytokine contributing to both physiological and pathological processes. Its superfamily members include TNF beta (TNF-β), CD40 ligand (CD40L), Fas ligand (FasL), TNF-related apoptosis-inducing ligand (TRAIL), and TNF superfamily member 14 (LIGHT) [12]. As a pro-inflammatory cytokine, TNF-α is one of the main mediators of inflammation-associated carcinogenesis in the TME. It exerts diverse effects through two receptors: TNF receptor type 1 (TNFR1) and TNF receptor type 2 (TNFR2). The major role of TNFR1 is the initiation of cell apoptosis, but the activation of TNFR1 can also induce cell survival mechanisms [13,14]. The determination of the final activity of TNFR1 is based on the concentration of TNF-α in the TME, as well as the effects of other involved cytokines [13]. As for TNFR2, its activation assists in regulating the immune and inflammatory responses, which can promote tumor growth and progression [13,14]. TNF-α is reported to have the potential to act as a diagnostic biomarker in oncological diseases [15,16,17,18,19]. In addition, previous studies have demonstrated that the increased expression of TNF-α is associated with poor survival in endometrial, colorectal, and hepatocellular cancers [17,19,20]. These findings indicate that TNF-α signaling may modulate the course of cancer, which could be therapeutically targeted to ameliorate clinical outcomes.

Because of the growing incidence of cancer, approximately 80% of cancer patients require anesthesia during cancer care for diagnostic, therapeutic, or palliative procedures, and over 60% of cancer patients receive anesthesia for primary surgical resection [4]. Clearly, anesthesia has become a requisite component of surgery and even cancer treatment. Growing evidence has shown that various anesthetic agents or techniques and perioperative management have distinct effects on innate and cellular immunity, the enhancement of adrenergic-inflammatory responses, and the activation of cancer-promoting cellular signaling pathways [2,4,5,6]. Moreover, there are retrospective studies reporting significant differences in survival among clinical anesthetic settings in different types of cancer surgeries [21,22,23,24,25,26,27]. A meta-analysis also concludes that the choice of anesthesia may be associated with long-term cancer outcomes [28]. Therefore, the following review aims to summarize an up-to-date overview of the current experimental and clinical evidence of the anesthesia implications on cancers based on the insights of TNF-α release. It also aims to provide protective anesthesia strategies for improving patient survival in the future.

## 2. General Anesthesia

### 2.1. Volatile Anesthetics

Volatile anesthetics (VAs) are used extensively in surgical oncology. Numerous in vitro studies have investigated the effects of VAs on human cancer cell lines and demonstrated that VAs seem to enhance the malignant potential of tumor cells and accelerate cancer progression [29,30,31,32,33,34]. Similar results are also observed in animal models [29,30]. However, some studies report inconsistent findings due to varied experimental protocols, environments, and subject species [35,36,37,38,39]. In addition, VAs have been reported to modulate the innate as well as the adaptive immune responses [40,41]. As a result, the effects of VAs on tumorigenesis may depend on the balance between the cellular signaling pathways and the immunosuppressive microenvironment formation.

#### 2.1.1. Laboratory Research

Research on the TNF-α release and its impact on cancer cell biology with the exposure to VAs is lacking (Table 1). Using human colon cancer cell lines, the exposure of isoflurane was reported to enhance resistance against TRAIL (a member of the TNF-α superfamily)-induced apoptosis through mechanisms related to caveolin-1, which suggested that the exposure of isoflurane may inadvertently decrease sensitivity to postoperative chemotherapeutic agents and even aggravate oncological outcomes [42].

#### 2.1.2. Clinical Studies

During cancer surgery, the effect of VAs on the TNF-α release is still uncertain. Chen et al. conducted a clinical study, which demonstrated that sevoflurane anesthesia significantly improved intraoperative hemodynamics and reduced the serum level of TNF-α in elderly patients undergoing lobectomy for lung cancer, but the incidence of postoperative complications simultaneously increased compared to propofol-based total intravenous anesthesia (TIVA) [43]. Moreover, Mahmoud et al. found that isoflurane anesthesia decreased the alveolar and plasma concentrations of TNF-α associated with one-lung ventilation (OLV) during open thoracic cancer surgery and had better postoperative outcomes compared with propofol anesthesia [44]. However, Jin et al. reported that, compared with propofol-based TIVA, sevoflurane anesthesia in patients undergoing lung cancer surgery under OLV exacerbated the injury to pulmonary function during the perioperative period via the increased release of serum TNF-α, aggravated lung edema, and inhibited hypoxic pulmonary vasoconstriction [45]. Qiao et al. also demonstrated that elderly patients receiving sevoflurane anesthesia during esophageal cancer surgery had a higher incidence of postoperative cognitive dysfunction (POCD) and an elevated plasma concentration of TNF-α than those maintained on propofol anesthesia [46]. Additionally, sevoflurane anesthesia and propofol-based TIVA did not make a difference in serum TNF-α production in patients undergoing major colorectal surgery for cancer disease or inflammatory bowel disease [47]. Similarly, an in vitro analysis study revealed no obvious impacts on TNF-α release during different types of anesthesia for breast cancer surgery [48]. Due to conflicting results, further large-scale and high-quality randomized control trials are required to determine the significance of these observations and the impact on prognosis after cancer surgery (Table 2; Figure 2).

### 2.2. Propofol

Propofol, a commonly used intravenous anesthetic, is widely administered in clinical anesthetic practice for cancer surgery. It has been reported to exert anti-tumor effects through various mechanisms in different cancer types [49]. Previous in vitro studies have shown that propofol can not only suppress the aggressiveness of human cancer cells [50,51,52,53,54,55,56,57], but can also alter the resistance against chemotherapeutic agents [51,58,59,60,61]. Animal studies provide similar information on the effects of propofol on tumor growth [50,58,62,63,64]. Furthermore, propofol can preserve the integrity of host immune function and mitigate surgery-induced immunosuppression [6,49]. Although there are few experimental investigations showing discrepancies [65,66], propofol is generally regarded to beneficially affect cancer development.

#### 2.2.1. Laboratory Research

In mouse neuroblastoma cell lines, propofol pretreatment ameliorated oxygen-glucose deprivation/reperfusion-induced inflammation and inhibited TNF-α production by modulating the nuclear transcription factor κB (NF-κB) pathway, indicating that propofol may be considered as a potential therapeutic approach to reduce inflammation in neuroblastoma [67]. However, Yan et al. revealed that treatment with propofol during tumor resection in murine models of breast cancer exhibited similar effects on reducing T lymphocyte cell counts, upregulating serum TNF-α production, and promoting distant metastasis compared with sevoflurane, which implied that the two common anesthetics modulated cancer progression through their immunosuppressive activity [68]. Current laboratory data about the effect of propofol on the release of TNF-α in cancer cell biology are limited, and further investigations are required (Table 1).

#### 2.2.2. Clinical Studies

During cancer surgery, propofol anesthesia had an inhibitory effect on the release of TNF-α in esophageal cancer patients [46] and contradictory results in lung cancer patients compared to inhalation anesthesia [43,44,45]. Moreover, the effect of propofol anesthesia on the plasma concentration of TNF-α did not differ from that of sevoflurane anesthesia during colorectal and breast cancer surgery [47,48]. Thus, although propofol anesthesia is recommended as a better choice for cancer surgery, prospective randomized control trials are needed to clarify its influence on the TNF-α release and cancer outcomes in the future (Table 2; Figure 2).

### 2.3. Ketamine

Ketamine is an N-methyl-D-aspartate receptor antagonist clinically used for anesthetic and analgesic purposes, which may provide anti-inflammatory and immunomodulatory effects in cancer surgery. Based on previous in vitro research, it has been determined that ketamine possesses the capacity to attenuate the malignant potential of various cancer cells [69,70,71,72,73,74]. However, there are inconsistent findings surrounding the relationship between ketamine administration and tumor metastasis from in vivo studies [64,75,76]. Additionally, ketamine seems to negatively influence cell-mediated immunity, especially nature killer (NK) cell activity, in animal models [64,76] but does not exhibit direct immunomodulation in clinical investigations [77,78]. Accordingly, there is a lack of evidence to advocate for the perioperative use of ketamine to improve outcomes in cancer patients.

#### 2.3.1. Laboratory Research

Basic research on the impact of ketamine administration on TNF-α production in cancer cell biology is limited (Table 1). Using mouse neuroblastoma cell lines, the administration of ketamine provided an anti-inflammatory effect on the lipopolysaccharide (LPS)-induced production of TNF-α through the inhibition of NF-κB and inducible nitric oxide synthase pathways [79].

#### 2.3.2. Clinical Studies

Ali et al. showed that ketamine administration could significantly suppress serum TNF-α production, especially when given in repeated doses, in patients undergoing radical prostatectomy [80]. During the laparoscopic radical resection of colorectal cancer, the addition of a single subanesthetic dose of ketamine was also reported to decrease the plasma level of TNF-α, prevent brain injury, and improve postoperative fatigue syndrome [81]. Similarly, Ren et al. showed that ketamine exerted an inhibitory effect on the release of serum TNF-α and improved the postoperative anxiety and depression of colorectal cancer patients [82]. In contrast, Cho et al. reported that subanesthetic doses of ketamine as an adjunct to desflurane anesthesia during colorectal cancer surgery did not convey any favorable impacts on the postoperative plasma level of TNF-α, NK cell activity, or long-term prognosis [77]. With ketamine as an anesthetic adjuvant, TIVA with propofol and remifentanil also failed to provide a direct immunomodulatory effect on the release of serum TNF-α during minimally invasive robotic radical prostatectomies [78]. Although the anti-inflammatory effect of ketamine on the TNF-α release is predominated, the impact of ketamine administration in cancer surgery is still debated (Table 2; Figure 2).

### 2.4. Dexmedetomidine

Dexmedetomidine (DEX), a highly selective α-2 adrenergic receptor agonist, exerts sedative, hypnotic, anxiolytic, sympatholytic, and analgesic properties. Due to its opioid-sparing and hemodynamic stabilizing effects, DEX has become an essential component of multimodal analgesia in major oncological surgeries. DEX is reported to restrain biological behaviors in human cancer cell lines [83,84,85,86] and suppress tumor progression in animal models through various pathways [83,86]. Nevertheless, there are contradictory results observed [87,88,89,90]. Moreover, the immunomodulatory effects of DEX from α-2 adrenergic receptors on various immune cells have been increasingly recognized [91]. Taken together, the impacts of DEX administration on cancer biology and immunity remain inconclusive, and further laboratory and clinical studies are warranted in the future.

#### 2.4.1. Laboratory Research

In rat models of ovarian cancer, DEX treatment could effectively decrease the plasma concentration of TNF-α and enhance the immunity through various signaling pathways, which eventually inhibited tumor growth, invasion, and migration [92,93]. Shin et al. also showed that DEX infusion during ovarian cancer surgery suppressed postoperative levels of serum TNF-α and cortisol, sped up the recovery of NK cell activity, and lowered the tumor burden after surgery in mouse models of ovarian cancer xenograft [94]. In addition, using rat pheochromocytoma cell lines, DEX significantly decreased oxygen-glucose deprivation/reperfusion-induced inflammation and apoptosis and reduced the release of TNF-α [95]. Based on the available data, DEX seems to have beneficial effects on the production of TNF-α and tumor progression (Table 1).

#### 2.4.2. Clinical Studies

In a meta-analysis, Wang et al. reported that DEX administration attenuated perioperative inflammation with a decreased TNF-α production and maintained the integrity of the immune function, which may contribute to diminish postoperative complications and improve clinical outcomes in surgical patients [96]. Pretreatment with DEX before the induction of general anesthesia (GA) was reported to protect lung cancer patients against lung injury during OLV by upregulating the expression of heme oxygenase-1 and reducing the serum production of TNF-α and reactive oxygen species [97]. Compared with GA plus epidural anesthesia (EA), adjunctive DEX in GA provided better hemodynamic control, dismissed inflammatory response with a reduction in serum TNF-α, protected intestinal function, and shortened the length of hospital stay in elderly patients with a gastrointestinal tumor after laparoscopic radical gastrectomy [98]. Dong et al. demonstrated that DEX could effectively reduce the release of plasma TNF-α by downregulating the expression of NF-κB and ameliorate the impaired immune function in patients receiving radical resections of gastric cancer [99]. Furthermore, DEX infusion decreased the plasma TNF-α concentration and improved the occurrence of POCD in patients undergoing surgical resections of colon, gynecological, gastric, and esophageal cancer [100,101,102,103]. Patients administered with DEX in colorectal cancer surgery also had better cardiocerebral protection with a reduced expression level of serum TNF-α [104]. Similarly, DEX could significantly stabilize hemodynamics, reduce inflammation via the downregulation of serum TNF-α, and inhibit free radical generation, playing a vital role in brain protection in patients undergoing craniotomy resections of glioma [105]. Liu et al. conducted a retrospective analysis and showed that DEX anesthesia could effectively stabilize hemodynamics and reduce the release of serum TNF-α compared with midazolam anesthesia for patients undergoing radical resections of ovarian cancer [106].

In addition to administration during cancer surgery, DEX could be an adjuvant for postoperative analgesia in cancer patients. During the postoperative period of patients undergoing a combined thoracoscopic–laparoscopic esophagectomy, the addition of DEX to sufentanil in intravenous patient-controlled analgesia (IVPCA) exhibited better postoperative analgesia, fewer inflammatory responses with a reduced plasma TNF-α level, and lower postoperative delirium categories and better health statuses [107]. Likewise, a combination of the intraoperative usage of DEX and postoperative IVPCA of DEX and ketorolac could provide adequate postoperative analgesia, reduce opioid consumption and relative complications, and alleviate immunosuppression and inflammation with a decrease in the serum TNF-α level in patients with lung cancer undergoing thoracoscopic surgery [108]. Song et al. also reported that DEX combined with oxycodone in IVPCA had better sedative and analgesic effects with fewer adverse reactions and reduced the release of serum TNF-α compared with oxycodone alone for patients with laparoscopic radical resections of rectal cancer [109]. In summation, perioperative DEX treatment seemed to have a beneficial effect on anti-inflammation by reducing the expression of TNF-α and improved postoperative outcomes in patients receiving cancer surgery (Table 2; Figure 2).

### 2.5. Systemic Lidocaine

Lidocaine is an amide local anesthetic and possesses an analgesic effect when used intravenously in perioperative settings. The administration of intravenous lidocaine during anesthesia has been associated with lowering the use of opioids, reducing the incidence of postoperative nausea and vomiting, and enhancing patients’ postoperative recovery [110]. Like other adjunctive agents, lidocaine has an inhibitory effect on carcinogenesis in experimental observations [111,112,113,114,115]. Apart from affecting cancer cell behavior, lidocaine can exhibit beneficial effects on components of inflammatory and immune responses [116]. Although results from in vitro and in vivo studies indicate that lidocaine may prevent against tumor progression, whether lidocaine can improve outcomes of surgical cancer patients needs to be investigated in prospective trials.

#### 2.5.1. Laboratory Research

Laboratory research on the impact of lidocaine treatment on the release of TNF-α in cancer cell biology is limited (Table 1). Ferreira et al. demonstrated that lidocaine in vitro and in vivo provided an anti-proliferative effect on tumor cells without affecting the release of TNF-α [117].

#### 2.5.2. Clinical Studies

During supratentorial tumor resections, systemic lidocaine administration improved early recovery quality and had neuroprotective effects, which may be attributed to its analgesic and inflammation-alleviating properties with a reduction in serum TNF-α [118]. Systemic lidocaine was also reported to improve postoperative recovery, alleviate inflammation and immunosuppression with a decrease in plasma TNF-α, and accelerate the return of bowel function in patients with gastric cancer undergoing laparoscopic radical gastrectomies [119]. Nevertheless, Herroeder et al. suggested that systemic lidocaine could significantly accelerate the return of bowel function and shorten the length of hospital stay after colorectal surgery by modulating pro-inflammatory cytokines other than TNF-α [120]. Accordingly, further investigations about the effects of systemic lidocaine on inflammatory responses and outcomes in patients receiving cancer surgery are needed (Table 2; Figure 2).

### 2.6. Midazolam

Midazolam is a common benzodiazepine medication, which can be used intravenously as an anesthetic for conscious sedation or as an adjunct for general and regional anesthesia. It has a rapid onset and short duration of action, and it causes relatively mild cardiovascular and respiratory effects. Previous in vitro and in vivo studies have shown an anti-tumorigenic property of midazolam [90,121,122,123,124]. However, midazolam is reported to provide distinct effects on various types of innate immune cells [63,124,125]. Hence, further findings from prospective studies are necessary before valid conclusions about the effects of perioperative midazolam use on the prognosis of cancer patients can be made.

#### 2.6.1. Laboratory Research

Basic research on the impact of midazolam on the release of TNF-α in cancer cell biology is limited (Table 1). Kang et al. conducted an in vitro and in vivo study and reported that midazolam could reduce the progression of hepatocellular carcinoma by inhibiting the NF-κB pathway and influence the immune microenvironment with a decrease in the secretion of TNF-α [124]. However, midazolam inhibited the growth of murine myeloid leukemia cell lines with an upregulated expression of TNF-α in a dose-dependent manner [126].

#### 2.6.2. Clinical Studies

Anesthetic induction with a higher dosage of midazolam in patients undergoing thoracoscopic resections of lung cancer was reported to provide better inhibition of the production of serum TNF-α and remarkably decrease the requirement for anesthetics, which may stabilize the hemodynamics of patients in the perioperative period and mitigate the severity of postoperative pain [127]. However, a retrospective analysis revealed that midazolam anesthesia was associated with less stable hemodynamics and an increased level of serum TNF-α than DEX anesthesia in patients undergoing radical resections of ovarian cancer [106]. Therefore, the role of midazolam in the TNF-α release and cancer progression is currently questionable and further investigations should be conducted (Table 2; Figure 2).

### 2.7. Thiopental

Thiopental is a rapid-onset and short-acting barbiturate agent, which is widely used in the treatment of seizure disorders and the induction of GA. Experimental studies have demonstrated that thiopental can enhance the aggressiveness of cancer cells [64,75]. Notably, thiopental affects immune cells to different extents. Thiopental is reported to suppress NK cell activity and impair neutrophil function [64,125] while protecting T lymphocytes from apoptosis [128]. Overall, we lack the evidence required to make recommendations about the perioperative use of thiopental in improving the prognosis of surgical cancer patients.

#### 2.7.1. Laboratory Research

Laboratory research on the impact of thiopental administration on TNF-α production in cancer cell biology is limited (Table 1). Using LPS-treated human glioma cell lines and murine brain inflammation models, thiopental was reported to exert anti-inflammatory effects with the suppression of TNF-α production by inhibiting NF-κB pathway activation [129].

#### 2.7.2. Clinical Studies

To the best of our knowledge, there is no clinical study on the relationship among thiopental, TNF-α, and cancer outcomes, which may contribute to the limited use of thiopental for anesthetic induction (Table 2).

### 2.8. Opioids

Opioids are often used in cancer patients for acute perioperative and chronic pain management. It is believed that opioids exert their influence on tumor growth and progression, but the role of opioids in cancer dissemination is conflicting [130,131,132,133,134,135,136]. Moreover, there is evidence that not all opioids have the same effect on immunity [137]. Previous prospective trials have failed to detect any correlation between opioid prescription and cancer progression [138,139]. However, results from retrospective studies exhibit a negative association between opioid exposure and survival [140,141,142]. The current state of knowledge suggests that pain is related to poor outcomes in cancer patients, but opioids should be judiciously prescribed due to their potential impact on cancer development [143]. Further prospective studies are needed to clarify possible effects of perioperative opioid administration on cancer prognosis.

#### 2.8.1. Laboratory Research

Using human cancer cell lines, morphine could dose-dependently suppress tumor cell growth via the inhibition of TNF-α release [144], which may be associated with reduced NF-κB pathway activation [145]. Fentanyl administration in human astrocytoma cell lines was also shown to inhibit TNF-α-induced chemokine expression, which may thereby be protective in response to inflammation-mediated neuropathogenesis [146]. However, Bastami et al. conducted an in vitro study and reported that various opioids had diverse influences on the release of TNF-α in LPS-treated human histiocytic lymphoma cell lines, in which tramadol and morphine had an inhibitory effect on TNF-α release and fentanyl had no effect [147]. In mouse models of breast cancer, tramadol attenuated tumor growth and hormone receptor expression, reduced the serum level of TNF-α, and preserved the NK cell activity compared with morphine [148]. Interestingly, Ma et al. declared that, when using mouse models of colorectal cancer, low-dose naltrexone (an opioid antagonist) decreased proliferation and promoted the apoptosis of cancer cells with increased TNF-α expression [149]. Results from laboratory research about the effects of opioids on the TNF-α release and cancer cell biology are inconclusive, and further investigations are required (Table 1).

#### 2.8.2. Clinical Studies

Compared with fentanyl, remifentanil administration during laparoscopic surgery for colon cancer was reported to significantly alleviate inflammatory responses with reduced secretion of serum TNF-α, improve oxidative stress indices, and decrease the frequency of adverse reactions [150]. In addition, oxycodone combined with flurbiprofen axetil applied for IVPCA could effectively reduce pain intensity, reverse immunosuppression, inhibit the release of serum TNF-α, and enhance postoperative intestinal recovery compared with sufentanil combined with flurbiprofen axetil in patients undergoing radical resections of colorectal cancer [151]. During thoracoscopic lobectomies for lung cancer, nalbuphine treatment prior to the induction of anesthesia also provided a significant analgesic effect, reduced the incidence of adverse reactions, and alleviated postoperative inflammatory responses with a reduced serum level of TNF-α [152]. However, Titon et al. demonstrated that opioid-free anesthesia may influence acute inflammation through a reduction of plasma interleukin (IL)-12 rather than TNF-α and other pro-inflammatory cytokines in the perioperative period of patients undergoing oncological surgery [153]. In summary, whether opioids possess an immunomodulatory capability and whether different types of opioids exert distinct effects on the release of TNF-α and subsequent cancer outcomes have been inconclusive thus far (Table 2; Figure 2).

### 2.9. Nonsteroidal Anti-Inflammatory Drugs

Nonsteroidal anti-inflammatory drugs (NSAIDs) play an essential role in multimodal analgesia during the perioperative period, and their main mechanism of action is the inhibition of cyclooxygenase. NSAIDs provide inhibitory effects on the development, growth, and invasion of human cancer cells [154,155,156,157]. The influence of NSAID treatment on inflammatory TME and cancer immunity is also becoming clearer [6,158]. However, retrospective data have shown an uncertain relationship between the perioperative use of NSAIDs and oncological outcomes in patients undergoing different types of cancer surgery [159,160,161]. Therefore, the current evidence is insufficient to encourage the routine use of NSAIDs in surgical cancer patients, and further prospective studies should be conducted.

#### 2.9.1. Laboratory Research

NSAID administration in vitro was reported to enhance TNF-α-mediated and TRAIL-induced cell death in different cancer cells through various mechanisms [162,163,164,165,166,167], of which aspirin and ibuprofen may be the least potent, while celecoxib and tamoxifen may be the most potent anti-inflammatory and anti-proliferative agents [168]. In addition, NSAIDs effectively inhibited the TNF-α-induced upregulation of cancer cell migration [166,169]. Setia et al. revealed that celecoxib administration prevented colitis-mediated colon carcinogenesis through the downregulation of inflammation with a decreased release of TNF-α and inactivation of the NF-κB pathway in mouse models [170]. NSAIDs in vivo could also reduce tumor growth via TNF-α-mediated and TRAIL-induced apoptosis [162,165,167,171]. In contrast, Vaish et al. reported that NSAIDs exerted anti-inflammatory and anti-neoplastic actions to reverse inflammation and carcinogenesis by upregulating the expression of TNF-α in rat models of colorectal cancer [172]. Moreover, Axiak-Bechtel et al. conducted a dog model of osteosarcoma and declared that NSAID use did not affect the stimulated production of TNF-α [173]. Based on the published data, further investigations are required to determine the effects of NSAIDs on the TNF-α release and cancer cell biology (Table 1).

#### 2.9.2. Clinical Studies

In clinical cancer surgical settings, the effect of perioperative NSAID prescription for pain management on the release of TNF-α is not fully discussed. Wen et al. demonstrated that postoperative analgesia with flurbiprofen axetil combined with fentanyl provided similar analgesic effects compared with fentanyl alone but significantly decreased the serum concentration of TNF-α in female patients undergoing breast cancer surgery [174]. Notably, preincisional parecoxib administration, compared with postincisional administration, was reported to reduce postoperative morphine consumption and attenuate serum IL-6 production rather than TNF-α after surgery but did not affect morphine-related adverse effects for patients undergoing colorectal cancer surgery [175]. Overall, NSAIDs seem to have beneficial effects on cancer progression through their anti-inflammatory actions, but further research discussing the relationship among NSAIDs, TNF-α, and cancer prognosis should be executed (Table 2; Figure 2).

## 3. Regional Anesthesia/Analgesia

### 3.1. Regional Anesthesia/Analgesia

Regional anesthesia/analgesia (RA), used either alone or in combination with GA, can reduce perioperative pain as well as opioid requirements and attenuate surgery-induced stress responses [4]. The regional techniques include the neuraxial block (EA or spinal anesthesia [SA]) and peripheral nerve block. Previous in vitro studies have shown a beneficial effect of RA in inhibiting the malignant potential of tumor cells [176,177]. Furthermore, the anti-inflammatory and analgesic effects of RA suggest a theoretical framework in which the perioperative immunity is optimally preserved [137]. There are observational studies showing better outcomes in patients receiving RA during cancer surgery [178,179], but prospective studies and meta-analyses reveal no significant benefit of RA in oncological outcomes [138,180,181,182,183,184]. Thus, RA may be reasonable for optimizing cancer patients’ comfort, but it does not seem to improve their outcomes based on controversial evidence.

#### 3.1.1. Laboratory Research

Research on the effect of RA on the release of TNF-α and cancer outcomes in experimental animal models is limited (Table 1). Inoue et al. recently conducted an in vivo study, which reported that GA combined with SA may decrease the total number of circulating tumor cells and reduce the stress response to surgery with a decreased level of serum TNF-α compared with GA alone in mouse models of prostate cancer [185].

#### 3.1.2. Clinical Studies

The thoracic paravertebral block was associated with lower incidence of postoperative delirium in elderly lung cancer patients undergoing thoracoscopic lobectomies compared with IVPCA, which may result from its opioid-sparing and anti-neuroinflammatory effects with a decreased surgery-induced production of TNF-α and neurofilament light [186]. Furthermore, the continuous wound infusion of local anesthetics significantly alleviated systemic inflammation with a reduced level of plasma TNF-α, decreased pain scores and opioid intake, and accelerated the recovery of respiratory function in patients of lung cancer resections [187]. During gastric cancer surgery, EA combined with GA could also preserve innate tumor immunity and mitigate stress responses with a reduced plasma concentration of TNF-α compared with GA alone [188,189]. In elderly patients undergoing liver cancer surgery, the addition of EA in GA similarly improved postoperative recovery and cognitive function and relieved inflammatory responses with a lower plasma level of TNF-α [190]. Recently, Geng et al. reported that a combination of a pectoral nerve block and stellate ganglion block effectively blunted perioperative inflammatory responses with a decreased level of serum TNF-α, alleviated acute postoperative pain, stabilized perioperative hemodynamics, and provided better postoperative sleep quality than a pectoral nerve block alone in breast cancer patients undergoing modified radical mastectomy [191]. Nevertheless, Okuda et al. demonstrated that thoracic EA during lung cancer surgery with OLV could attenuate local inflammation by decreasing the production of IL-6 rather than TNF-α [192]. A combination of thoracic paravertebral block or EA and GA was also reported to decrease pain scores, reduce the use of opioids and vasoactive agents, and improve perioperative immune function and long-term outcomes in patients undergoing esophageal cancer surgery, but it did not affect the release of serum TNF-α [193]. Karadeniz et al. similarly reported that GA combined with EA plus patient-controlled epidural analgesia provided lower intraoperative opioid consumption and shorter hospital stay in comparison with GA plus IVPCA in bladder cancer patients undergoing radical cystectomy, but it had no effect on the serum level of TNF-α [194]. Additionally, during radical prostatectomies, thoracic EA could significantly blunt the early neuroendocrine response to surgery by preventing an increase in plasma cortisol and glucose levels and reducing postoperative pain without affecting the secretion of plasma TNF-α [195]. Notably, Siekmann et al. reported that surgical and analgesic techniques had no impact on the postoperative release of plasma TNF-α following colorectal cancer surgery [196]. In summary, RA seems to provide beneficial effects on surgical cancer patients, but further prospective trials are needed to investigate the impacts of RA on cancer progression by modulating perioperative TNF-α release (Table 2; Figure 2).

## 4. Perioperative Care

### 4.1. Body Temperature

Body temperature control is crucial for optimal metabolism and physiological processes, especially during surgery. Perioperative hypothermia may lead to detrimental complications and contribute to perioperative morbidity and mortality [197]. In addition, perioperative hypothermia is reported to exacerbate surgery-induced immunosuppression, which may worsen prognoses in cancer patients [198,199,200]. However, an in vivo study has demonstrated that hypothermia appears to have no significant effects on NK activity and resultant tumor metastasis [64]. Although there is no sufficient evidence elucidating the influence of hypothermia on cancer progression, maintaining normothermia during cancer surgery is still strongly recommended.

#### 4.1.1. Laboratory Research

The effect of body temperature on the TNF-α release in cancer cell biology has been the subject of debate (Table 1). Du et al. reported that hypothermia in vitro and in vivo activated adipocytes to stimulate lung cancer progression by increasing the release of TNF-α [201]. However, mild hyperthermia significantly enhanced the efficacy of immunotherapy and reduced the risk of cancer metastasis in mouse models of pancreatic cancer, which may contribute to increasing drug accumulation and improving the anti-tumor immune activity with increased secretion of TNF-α [202].

#### 4.1.2. Clinical Studies

Preoperative hyperthermia could improve the immune responses to surgical stress by decreasing the plasma concentration of TNF-α without enhancing the quality of recovery after colorectal cancer surgery [203]. In contrast, Atanackovic et al. found that performing hyperthermia showed a redistribution of innate immune cells with an increase in serum TNF-α in patients with solid cancers [204]. During cytoreductive surgery and hyperthermic intraperitoneal chemotherapy, Coccolini et al. reported that serum and peritoneal concentrations of TNF-α did not change significantly throughout the whole procedure [205]. Furthermore, warmed and humidified carbon dioxide (CO_2_) insufflation was more effective than standard CO_2_ insufflation in maintaining patients’ heat homeostasis, but it showed no benefit in terms of pain scores and plasma TNF-α and other cytokine levels during robotic radical prostatectomies [206]. As for hypothermia, Hansen et al. did not detect a significant difference in the serum level of TNF-α in patients with perioperative moderate hypothermia scheduled for the resection of malignant melanoma [207]. Consequently, although maintaining normothermia during cancer surgery is recommended, the effect of body temperature control on the release of TNF-α and cancer outcomes warrants further investigations (Table 2; Figure 2).

### 4.2. Hyperglycemia

Hyperglycemia is associated with increased postoperative complications in different types of surgery, including oncological resections. Perioperative hyperglycemia can enhance surgery-induced immunosuppression, which may increase patients’ vulnerability to postoperative wound infection and potential tumor dissemination [199,208]. Moreover, the hyperglycemic condition can induce the activation of signaling pathways in cancer progression and confer resistance against chemotherapy [209]. As a result, perioperative hyperglycemia in surgical cancer patients should be avoided, and further studies are warranted to confirm the association between perioperative hyperglycemia and tumor spread or metastasis.

#### 4.2.1. Laboratory Research

Laboratory research on the impact of hyperglycemia on TNF-α production in cancer cell biology is limited (Table 1). Otto et al. conducted an in vitro study and showed that the interaction between hyperglycemia and macrophages could promote the expression of TNF-α, as well as epithelial–mesenchymal transition transcription factors in pancreatic ductal epithelial cells, which may aggravate malignancy-associated alterations [210].

**Table 1 cancers-15-00739-t001:** Experimental studies on the effects of anesthetics/analgesics and perioperative care on TNF-α release and cancer cell biology.

Anesthetics/Analgesics and Perioperative Management	TNF-α Release	Effects
**VAs**	Not applicable	There is no relevant experimental study.
**Propofol**	Decreased	Propofol ameliorated oxygen-glucose deprivation/reperfusion-induced inflammation in mouse neuroblastoma cell lines [67].
	Increased	Propofol suppressed the proliferation and killing activity of anti-tumor immune cells during tumor resection in murine models of breast cancer [68].
**Ketamine**	Decreased	Ketamine attenuated LPS-induced inflammatory responses in mouse neuroblastoma cell lines [79].
**DEX**	Decreased	DEX improved the immune function and decreased the tumor invasion and migration in rat models of ovarian cancer [92]; DEX enhanced the immune function and suppressed the tumor growth in rat models of ovarian cancer [93]; DEX suppressed surgical stress responses, sped up the recovery of NK cell activity, and lowered the tumor burden after surgery in mouse models of ovarian cancer [94]; DEX attenuated oxygen-glucose deprivation/reperfusion-induced inflammation and apoptosis in rat pheochromocytoma cell lines [95].
**Lidocaine**	No difference	Lidocaine contributed to tumor reduction but had no influence on tumor-induced inflammatory responses [117].
**Midazolam**	Decreased	Midazolam reduced the progression of hepatocellular carcinoma cell lines and influenced the immune microenvironment in mouse models [124].
	Increased	Midazolam inhibited the growth and induced the differentiation of murine myeloid leukemia cell lines [126].
**Thiopental**	Decreased	Thiopental exerted anti-inflammatory effects on LPS-treated human glioma cell lines and murine brain inflammation models [129].
**Opioids**	Decreased	Morphine dose-dependently suppressed tumor cell growth in human cancer cell lines [144]; morphine and its derivatives induced the apoptosis of human cancer cell lines [145]; tramadol and morphine inhibited cytokine release in LPS-treated human histiocytic lymphoma cell lines [147].
	Increased	Naltrexone decreased proliferation and promoted apoptosis in mouse models of colorectal cancer [149].
	No difference	Fentanyl had no effect on the production of cytokines in LPS-treated human histiocytic lymphoma cell lines [147].
**NSAIDs**	Decreased	Celecoxib downregulated inflammation and prevented against colitis-mediated colon carcinogenesis in mouse models [170].
	Increased	Sulindac and celecoxib exerted anti-inflammatory and anti-neoplastic actions in rat models of colorectal cancer [172].
	No difference	NSAIDs had no influence on cytokine release in dog models of osteosarcoma [173].
**RA**	Decreased	The combination of GA and SA reduced stress responses to surgery and attenuated the suppression of innate tumor immunity compared with GA alone in mouse models of prostate cancer [185].
**Body temperature**	Increased	Hypothermia stimulated lung cancer boost [201]; mild hyperthermia enhanced the efficacy of immunotherapy and reduced the risk of cancer metastasis in mouse models of pancreatic cancer [202].
**Hyperglycemia**	Increased	The interaction between hyperglycemia and macrophages promoted malignancy-associated alterations in pancreatic ductal epithelial cells [210].
**Blood transfusion**	Not applicable	There is no relevant experimental study.

DEX = dexmedetomidine; GA = general anesthesia; LPS = lipopolysaccharide; NK cell = nature killer cell; NSAID = non-steroidal anti-inflammatory drug; SA = spinal anesthesia; TNF-α = tumor necrosis factor alpha; VA = volatile anesthetic; RA = regional anesthesia/analgesia.

#### 4.2.2. Clinical Studies

The degree of perioperative glucose fluctuation was positively associated with the postoperative level of serum TNF-α and poor prognosis in patients with supratentorial neoplasms undergoing intracranial excision [211]. Notably, perioperative hyperglycemia was shown to attenuate postoperative immune activation with a significant suppression of serum TNF-α release in cancer patients of esophageal or pancreatic resections [212]. Collectively, the limited data show that the glucose fluctuation and hyperglycemia may affect the TNF-α expression and subsequent cancer progression, which should be clarified through further investigations (Table 2; Figure 2).

### 4.3. Blood Transfusion

Blood transfusion is a commonly used adjunctive therapy for surgical diseases. Cancer patients often suffer from anemia prior to operation and require a blood transfusion during the perioperative period. However, blood transfusion can modulate the immune responses and may increase the incidence of cancer recurrence and postoperative infection [198,199]. Available clinical data summarized in meta-analyses show that blood transfusion seems to enhance the risks of postoperative complications, cancer recurrence, and worse patient survival [213,214,215,216]. Taken together, although more aggressive types of cancer may need more requirements for blood transfusion during surgery, using a restrictive threshold of blood transfusion in cancer patients should be mandatory.

#### 4.3.1. Laboratory Research

To the best of our knowledge, there is no experimental study about the relationship among blood transfusion, TNF-α release, and cancer cell biology (Table 1).

#### 4.3.2. Clinical Studies

During colorectal cancer surgery, patients who received an allogenic red blood cell (RBC) transfusion had a significantly increased concentration of plasma TNF-α after surgery compared to those who did not [217]. Surinenaite et al. also showed that the serum concentration of TNF-α was increased in RBC-transfused patients with advanced colorectal cancer after surgical treatment but did not significantly change in those with early colorectal cancer [218]. Compared with homologous RBC transfusion, autologous RBC transfusion was reported to alleviate transfusion-related immunosuppression by increasing the plasma TNF-α level in esophageal and colorectal cancer patients undergoing radical resections [219,220]. In addition, Geng et al. reported that, compared with allogeneic suspended RBC transfusion, allogeneic leukocyte-depleted RBC transfusion could effectively alleviate inflammation with a less decreased level of serum TNF-α, improve coagulation function, reduce stress response, and enhance wound healing without increasing adverse reactions or the postoperative infection rate in patients with recurrences of colon cancer after operation [221]. However, Benson et al. revealed that, although stored leukocyte-depleted or non-leukocyte-depleted packed RBC both contained tumorigenic mediators that may enhance tumor progression, their TNF-α levels did not differ [222]. Beck-Schimmer et al. similarly demonstrated that the retransfusion of irradiated intraoperative cell salvage blood did not upregulate the concentrations of serum TNF-α and other inflammatory mediators in patients undergoing gynecological cancer surgery [223]. Interestingly, a low plasma TNF-α level before surgery was associated with a high perioperative transfusion rate in glioma surgical patients, which may indicate that the preoperative immune response could influence perioperative transfusion requirements [224]. In summation, according to the available data, perioperative RBC transfusion seems to influence the release of TNF-α and even cancer prognosis; however, further prospective trials are needed to build definitive evidence (Table 2; Figure 2).

**Table 2 cancers-15-00739-t002:** Clinical studies on the effects of anesthetics/analgesics and perioperative care on TNF-α release and clinical outcomes.

Anesthetics/Analgesics and Perioperative Management	TNF-α Release	Effects
**VAs**	Decreased	Sevoflurane improved hemodynamics and inflammatory responses in lung lobectomies but increased the incidence of postoperative complications compared with propofol [43]; isoflurane decreased inflammatory responses associated with OLV during open thoracic cancer surgery and had better postoperative outcomes compared with propofol [44].
	Increased	Sevoflurane exacerbated the injury to pulmonary function in lung cancer surgery compared with propofol [45]; sevoflurane increased the incidence of POCD after esophageal cancer surgery compared with propofol [46].
	No difference	Sevoflurane had no effect on complement activation and cytokine release in major colorectal cancer surgery compared with propofol [47]; sevoflurane had no effect on NK cell and cytotoxic T lymphocyte counts or the apoptosis rate in breast cancer surgery compared with propofol [48].
**Propofol**	Decreased	Propofol attenuated the injury to pulmonary function in lung cancer surgery compared with sevoflurane [45]; propofol decreased the incidence of POCD after esophageal cancer surgery compared with sevoflurane [46].
	Increased	Propofol enhanced unstable hemodynamics and inflammatory responses in lung lobectomies but decreased the incidence of postoperative complications compared with sevoflurane [43]; propofol increased inflammatory responses associated with OLV during open thoracic cancer surgery and had worse postoperative outcomes compared with isoflurane [44].
	No difference	Propofol had no effect on complement activation and cytokine release in major colorectal surgery compared with sevoflurane [47]; propofol had no effect on NK cell and cytotoxic T lymphocyte counts or the apoptosis rate in breast cancer surgery compared with sevoflurane [48].
**Ketamine**	Decreased	Ketamine suppressed pro-inflammatory cytokine production in radical prostatectomies [80]; ketamine prevented brain injury and improved postoperative fatigue syndrome after laparoscopic colorectal cancer surgery [81]; ketamine reduced inflammatory responses and improved postoperative anxiety and depression after colorectal cancer surgery [82].
	No difference	Ketamine in colorectal cancer surgery had no impact on postoperative NK cell activity, inflammatory responses, or long-term prognosis [77]; ketamine had no direct immunomodulation during minimally invasive robotic radical prostatectomies [78].
**DEX**	Decreased	DEX reduced the oxidative stress and inflammation in lung lobectomies with OLV [97]; DEX in GA provided better hemodynamic control, dismissed inflammatory responses, protected intestinal function, and shortened the length of hospital stay after laparoscopic radical gastrectomies compared with GA plus EA [98]; DEX attenuated systemic inflammation and ameliorated the impaired immune function in radical gastric cancer surgery [99]; DEX improved postoperative cognitive function after laparoscopic colon cancer surgery [100]; DEX reduced adverse responses and the occurrence of POCD after laparoscopic total hysterectomies [101]; DEX reduced postoperative inflammation and promoted the recovery of postoperative cognitive function after gastric cancer surgery [102]; DEX, during esophageal cancer surgery, alleviated the incidence of POCD [103]; DEX provided better cardiocerebral protection in colorectal cancer surgery [104]; DEX stabilized hemodynamics, reduced inflammation, and inhibited free radical generation in glioma surgery [105]; DEX stabilized hemodynamics and alleviated stress responses in ovarian cancer surgery compared with midazolam [106].
**Lidocaine**	Decreased	Lidocaine, during supratentorial tumor resections, improved early recovery quality and had brain-injury alleviation effects [118]; lidocaine improved postoperative recovery, alleviated inflammation and immunosuppression, and accelerated the return of bowel function after laparoscopic radical gastrectomies [119].
	No difference	Lidocaine accelerated the return of bowel function and shortened the length of hospital stay after colorectal surgery [120].
**Midazolam**	Decreased	Midazolam, during thoracoscopic lung cancer surgery, inhibited inflammatory responses, decreased the requirement of anesthetics, stabilized perioperative hemodynamics, and mitigated postoperative pain [127].
	Increased	Midazolam enhanced unstable hemodynamics and stress responses in ovarian cancer surgery compared with DEX [106].
**Thiopental**	Not applicable	There is no relevant clinical study.
**Opioids**	Decreased	Nalbuphine enhanced analgesic effects, reduced the incidence of adverse reactions, and alleviated postoperative inflammatory responses in thoracoscopic lung lobectomies [152].
	No difference	Opioid-free anesthesia influenced acute inflammation in the perioperative period of oncological surgery [153].
**NSAIDs**	Decreased	Flurbiprofen axetil plus fentanyl provided similar postoperative analgesia in breast cancer surgery compared with fentanyl alone [174].
**RA**	Decreased	Thoracic paravertebral blocks decreased the incidence of postoperative delirium and enhanced pain control and recovery quality compared with IVPCA in thoracoscopic lung lobectomies [186]; continuous wound analgesia alleviated systemic inflammation, decreased pain scores and opioid intake, and accelerated the recovery of respiratory function in lung cancer surgery [187]; EA, during gastric cancer surgery, mitigated inflammatory responses, preserved innate tumor immunity, and decreased the incidence of postoperative adverse reactions [188]; EA, during gastric cancer surgery, reduced the stress reactions and maintained the integrity of immune function [189]; EA improved postoperative recovery and cognitive function and relieved inflammatory responses in liver cancer surgery [190].
	No difference	Thoracic EA attenuated local inflammation but did not affect systemic inflammation during lung cancer surgery [192]; thoracic paravertebral blocks or EA decreased pain scores, reduced the use of opioids and vasoactive agents, and improved perioperative immune function and postoperative survival after esophageal cancer surgery [193]; GA plus EA and patient-controlled epidural analgesia provided lower opioid consumption and shorter hospital stay in radical cystectomies compared with GA plus IVPCA [194]; thoracic EA blunted early postoperative stress responses and reduced postoperative pain after radical prostatectomies [195]; surgical rather than analgesic techniques had a great impact on postoperative inflammation in colorectal cancer surgery [196].
**Body temperature**	Decreased	Preoperative hyperthermia improved immune responses to surgical stress but did not enhance the quality of recovery after colorectal cancer surgery [203].
	Increased	Cancer patients treated with hyperthermia had a redistribution of innate immune cells [204].
	No difference	The cytokine levels paralleled hemodynamic and metabolic derangements in the hyperthermic phase of cytoreductive surgery and hyperthermic intraperitoneal chemotherapy [205]; warmed and humidified CO_2_ insufflation had no benefit in terms of pain scores and cytokine levels during robotic radical prostatectomies [206]; moderate hypothermia during malignant melanoma resection affected the release of circulating cytokines and adhesion molecules [207].
**Hyperglycemia**	Decreased	Perioperative hyperglycemia attenuated postoperative immune activation during esophageal or pancreatic cancer resections [212].
	Increased	The degree of perioperative glucose fluctuation was positively related to postoperative cytokine levels and poor short-term prognosis after supratentorial tumor resection [211].
**Blood transfusion**	Increased	Allogenic RBC transfusion increased the serum levels of cytokines in colorectal cancer surgery [217]; RBC transfusion in advanced colorectal cancer surgery exacerbated the antioxidative and immune systems and increased serum cytokine levels [218].

CO_2_ = carbon dioxide; DEX = dexmedetomidine; EA = epidural anesthesia; GA = general anesthesia; IVPCA = intravenous patient-controlled analgesia; NK cell = nature killer cell; NSAID = non-steroidal anti-inflammatory drug; OLV = one-lung ventilation; POCD = postoperative cognitive dysfunction; TNF-α = tumor necrosis factor alpha; VA = volatile anesthetic; RA = regional anesthesia/analgesia; RBC = red blood cell.

## 5. TNF-α in Cancer

TNF-α is reported to be associated with inflammation-associated carcinogenesis. It is mainly secreted by immune cells such as monocytes, macrophages, NK cells, T lymphocytes, mast cells, and neutrophils, but can also be produced by non-immune cells such as endothelial cells, adipocytes, neurons, fibroblasts, and smooth muscle [14]. Additionally, TNF-α not only induces its own secretion, but also stimulates the production of other inflammatory cytokines and chemokines [12]. TNF-α and its receptors exert biological functions in cancer cells by activating distinct signaling pathways, including NF-κB and c-Jun N-terminal kinase (JNK), in which NF-κB is an anti-apoptotic signal while sustained JNK activation contributes to cell death [225]. The crosstalk between the NF-κB and JNK signaling is involved in determining cellular outcomes in response to TNF-α, which depends on cytokine concentrations. In high concentrations, TNF-α provides an anti-tumor action by mediating cellular apoptosis, directing tumor-associated macrophages to anti-tumoral phenotypes, guiding neutrophils and monocytes to tumor sites, activating macrophages and inhibiting monocyte differentiation to immunosuppressive phenotypes, and inducing the disruption of tumor vasculature; in contrast, TNF-α expression at low levels can be pro-tumorigenic [14].

The TME is a complex biological ecosystem of solid tumors. As mentioned above, anesthetics/analgesic techniques and perioperative management may affect the release of TNF-α, but there are inconsistent results. The potential causes include the following: first, TNF-α has been described as having dual effects on almost every type of cancer [14]; second, more detailed and complicated mechanisms about the effects of TNF-α on cancer cells beyond the NF-κB and JNK signaling pathways are increasingly elucidated [13,14]; third, perioperative anesthetic strategies may have impacts to different extents on various immune and non-immune cells that can produce TNF-α; and last, differences in clinical surgical settings, including cancer types, patient conditions, and surgical procedures, probably confound the study consequences. Due to these reasons, the effects of anesthesia on TNF-α release and resultant cancer outcomes eventually become minimal or none.

## 6. Conclusions

Cancer remains the leading cause of death worldwide, and surgical resection is currently the mainstay treatment for solid tumors. However, tumor recurrence and/or metastasis after surgical resection can lead to morbidities and mortalities in most cancer patients. During surgery, perioperative anesthetic strategies may alter the TME and even affect cancer outcomes. Herein, we endeavor to summarize the role of anesthesia in cancers from the insights of TNF-α release and subsequent cancer progression. The scarce evidence from the available data suggests that patients maintained on VA-based anesthesia or propofol-based TIVA during cancer surgery have no clear significance in the TNF-α release and patient outcomes, but adjunctive agents including ketamine, DEX, systemic lidocaine, and opioids appear to have a beneficial effect on the release of TNF-α and may have a tendency towards the suppression of carcinogenesis. Additionally, the addition of RA and the reduction of blood transfusion provide immunomodulatory effects in cancer patients undergoing surgical resections. Currently, it remains challenging to recommend specific anesthetics/analgesic techniques or perioperative management during cancer surgery for optimal risk reduction with regard to tumor recurrence or metastasis. Therefore, further in-depth experimental research and high-quality clinical trials are needed to explore the detailed molecular mechanisms involved in the relationship between anesthesia, TNF-α release, and cancer progression.

## Figures and Tables

**Figure 1 cancers-15-00739-f001:**
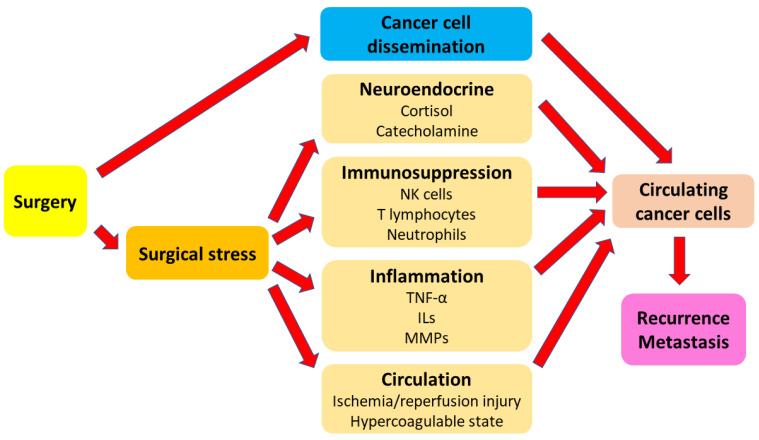
Overview of surgery and surgical stress response, which may jointly contribute to cancer recurrence and metastasis. IL = interleukin; MMP = matrix metalloproteinase; NK cell = nature killer cell; TNF-α = tumor necrosis factor alpha.

**Figure 2 cancers-15-00739-f002:**
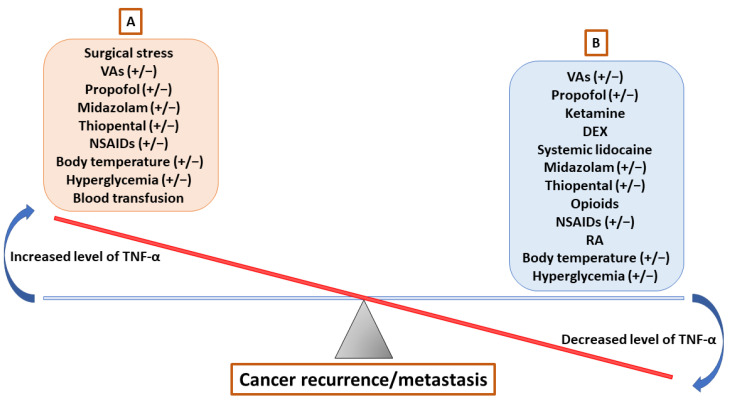
A theory of balance between TNF-α-promoting and -inhibiting factors contributing to cancer recurrence/metastasis. (**A**) Surgical stress and blood transfusion may promote cancer recurrence/metastasis by increasing the TNF-α level; (**B**) ketamine, DEX, lidocaine, opioids, and RA may reduce cancer recurrence/metastasis by decreasing the TNF-α release. DEX = dexmedetomidine; NSAID = non-steroidal anti-inflammatory drug; TNF-α = tumor necrosis factor alpha; VA = volatile anesthetic; RA = regional anesthesia/analgesia.

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
