# Peer review of "Tumor Necrosis Factor Alpha: Implications of Anesthesia on Cancers"

_cancers, 2023, doi:10.3390/cancers15030739_

Round 1

Reviewer 1 Report

Manuscript (Manuscript ID: cancers-2153228) titled: “Tumor Necrosis Factor: Molecular Insights and Clinical Implications of Anesthesia on Cancers”, has been reviewed. The authors provided a narrative review of the effect of mode of anesthesia and drugs on tumor necrosis factor and recurrence and metastasis of cancer. This review is comprehensive and the manuscript is well written. Moreover, the author has published many papers in this field and is an expert in this field. I only have 2 minor comments.

1. The interpretation of figure 1 is very conclusive and should be clearly presented. Surgery is an invasive medical treatment, and it is reasonable that surgery exerts a stress response. However, the role of anesthesia on stress response is not clear in this figure 1. Whether anesthesia reduce or promote stress response needs to be defined. The authors used totally 4 colors of arrows in (red, green, blue and grey) in this figure, however, which color represents promotion or inhibition needs to be explained. Besides, what does the term anesthesia in this figure refer to in general? If it is difficult to define clearly, this diagram will only add confusion to the readers. Suggest only keep surgery on stress responses in this figure.

2. Another concern is that table 2 is too lengthy and too complicated. It seems that the purpose of simplifying the concept of tabulation has been lost. Strongly recommended to simplify it. Suggest draw a conclusive figure to demonstrate different anesthetic on tumor necrosis factor and actual clinical outcomes.

Author Response

Original comments of reviewer 1:

1) The interpretation of figure 1 is very conclusive and should be clearly presented. Surgery is an invasive medical treatment, and it is reasonable that surgery exerts a stress response. However, the role of anesthesia on stress response is not clear in this figure 1. Whether anesthesia reduce or promote stress response needs to be defined. The authors totally used 4 colors of arrows in (red, green, blue and grey) in this figure, however, which color represents promotion or inhibition needs to be explained. Besides, what does the term anesthesia in this figure refer to in general? If it is difficult to define clearly, this diagram will only add confusion to the readers. Suggest only keep surgery on stress responses in this figure.

2) Another concern is that table 2 is too lengthy and too complicated. It seems that the purpose of simplifying the concept of tabulation has been lost. Strongly recommended to simplify it. Suggest draw a conclusive figure to demonstrate different anesthetic on tumor necrosis factor and actual clinical outcomes.

Reply by authors:

1) Thanks for your wise comments. We have made modification as your suggestion. Please see modified Figure 1.

2) Thanks for your wise comments. We have removed references with less relationship to simplify Table 2 as your suggestions. In addition, we have drawn Figure 2 to conclude current evidence on the effects of anesthesia on TNF-α release and potential oncological outcomes. Please see the description of modified Table 2 and Figure 2.

Reviewer 2 Report

The review by Tseng and colleagues report on the relationships between several anesthetic compounds/types and TNFalfa. Overall the review is highly interesting and a nice contribution to the field. There are very small mistakes in the text such as missing words such as "the" and "is" which can be easily corrected by the authors.

Author Response

Original comments of reviewer 2:

1) The review by Tseng and colleagues report on the relationships between several anesthetic compounds/types and TNF alpha. Overall the review is highly interesting and a nice contribution to the field. There are very small mistakes in the text such as missing words such as "the" and "is" which can be easily corrected by the authors.

Reply by authors:

1) Thanks for your comments. The manuscript has been carefully reviewed by an experienced editor whose first language is English and who specializes in editing papers written by scientists whose native language is not English. Please see the attached file.
